# Process Development for GMP-Grade Full Extract Cannabis Oil: Towards Standardized Medicinal Use

**DOI:** 10.3390/pharmaceutics17070848

**Published:** 2025-06-28

**Authors:** Maria do Céu Costa, Ana Patrícia Gomes, Iva Vinhas, Joana Rosa, Filipe Pereira, Sara Moniz, Elsa M. Gonçalves, Miguel Pestana, Mafalda Silva, Luís Monteiro Rodrigues, Anthony DeMeo, Logan Marynissen, António Marques da Costa, Patrícia Rijo, Michael Sassano

**Affiliations:** 1CBIOS—Universidade Lusófona’s Research Center for Biosciences & Health Technologies, Campo Grande 376, 1749-024 Lisboa, Portugal; pg@somaipharma.eu (A.P.G.); monteiro.rodrigues@ulusofona.pt (L.M.R.); 2NICiTeS, Polytechnic Institute of Lusophony, ERISA-Escola Superior de Saúde Ribeiro Sanches, Rua do Telhal aos Olivais 8, 1950-396 Lisboa, Portugal; 3SOMAÍ Pharmaceuticals, R. 13 de Maio 52, 2580-507 Carregado, Portugal; imv@somaipharma.eu (I.V.); jr@somaipharma.eu (J.R.); fjp@somaipharma.eu (F.P.); sjm@somaipharma.eu (S.M.); symbioticad@gmail.com (A.D.); lm@somaipharma.eu (L.M.); amc@somaipharma.eu (A.M.d.C.); ms@somaipharma.eu (M.S.); 4INIAV, Instituto Nacional de Investigação Agrária e Veterinária I.P., Quinta do Marquês, 2780-157 Oeiras, Portugal; elsa.goncalves@iniav.pt (E.M.G.); miguel.pestana@iniav.pt (M.P.); mafaldasdos@gmail.com (M.S.); 5GeoBiotec—GeoBioTec Research Institute, Faculdade de Ciências e Tecnologia, Universidade Nova de Lisboa, Campus da Caparica, 2829-516 Caparica, Portugal; 6Faculdade de Ciências e Tecnologia, Universidade Nova de Lisboa, Campus da Caparica, 2829-516 Caparica, Portugal

**Keywords:** medicinal cannabis, deep-cooled ethanol, extraction flow chart, sensory evaluation, full-spectrum cannabinoid extract, cannabinoids, terpenes, high-THC Δ^9^-tetrahydrocannabinol (THC)

## Abstract

**Background/Objectives:** The industrial extraction and purification processes of *Cannabis sativa* L. compounds are critical steps in creating formulations with reliable and reproducible therapeutic and sensorial attributes. **Methods:** For this study, standardized preparations of chemotype I were chemically analyzed, and the sensory attributes were studied to characterize the extraction and purification processes, ensuring the maximum retention of cannabinoids and minimization of other secondary metabolites. The industrial process used deep-cooled ethanol for selective extraction. **Results:** Taking into consideration that decarboxylation occurs in the process, the cannabinoid profile composition was preserved from the herbal substance to the herbal preparations, with wiped-film distillation under deep vacuum conditions below 0.2 mbar, as a final purification step. The profiles of the terpenes and cannabinoids in crude and purified Full-spectrum Extract Cannabis Oil (FECO) were analyzed at different stages to evaluate compositional changes that occurred throughout processing. Subjective intensity and acceptance ratings were received for taste, color, overall appearance, smell, and mouthfeel of FECO preparations. **Conclusions:** According to sensory analysis, purified FECO was more accepted than crude FECO, which had a stronger and more polarizing taste, and received higher ratings for color and overall acceptance. In contrast, a full cannabis extract in the market resulted in lower acceptance due to taste imbalance. The purification process effectively removed non-cannabinoids, improving sensory quality while maintaining therapeutic potency. Terpene markers of the flower were remarkably preserved in SOMAÍ’s preparations’ fingerprint, highlighting a major qualitative profile reproducibility and the opportunity for their previous separation and/or controlled reintroduction. The study underscores the importance of monitoring the extraction and purification processes to optimize the cannabinoid content and sensory characteristics in cannabis preparations.

## 1. Introduction

The knowledge on the composition of *Cannabis sativa* L. concerns nearly 600 identified chemical compounds, including more than 100 terpenes and over 150 phytocannabinoids, whose identification and quantification are growing as analytical methods also evolve, making it possible to understand the complexity of the various cannabinoids and their isomers [1]. The current focus of cannabis research is its potential medical applications, but little is known about the extracts representing the pharmacological profile of the sourced inflorescences.

The organoleptic characteristics of botanical extracts may influence therapeutic adherence. Although no scientific reference has been found that directly addresses the relationship between any unpleasant taste of *C. sativa* extracts and treatment adherence, it is recognized that the volatile organic compounds (primarily terpenes) and chlorophyll present can affect the flavor and aroma of cannabis products, potentially influencing the patient experience [2,3].

The resin glands (trichomes), terpenes, polyphenols, and even cannabinoids that cannabis produces may be considered a part of these plant defense mechanisms to avoid being eaten [4]. Some polyphenols may appear as protectants against stress factors [5], e.g., the cultivars infected by the Hop Latent Viroid (HLVd) disease will contain increased amounts of polyphenols, reflected in both the yield and loss of quality, including up to a 50% reduction in both cannabinoid and terpene production [6]. While the ecological functions of cannabinoids include protection against UV light and desiccation as well as plant defense, other chemicals may be produced from biotic or abiotic stressors for defense mechanisms under water stress, UV stress, mechanical stress, and pathogenic stress [7,8,9].

When processed, bitter compounds like sesquiterpene lactones and acids are solubilized during cannabis extraction, resulting in poor-tasting terpenoid content and cannabinoid ratios, which provide the most explicit demarcation between chemovars [10,11].

When a manufacturer produces a crude THC extract, independent of the extraction method, the resulting oil is composed of the acidic forms of cannabinoids. Crude extracts must undergo decarboxylation to remove the carboxylic acid from tetrahydrocannabinolic acid (THCA) and convert the molecules to their more psychoactive counterpart, Δ^9^-THC. To achieve decarboxylation, the oil must be exposed to high temperatures for a specific residence time to break the carbon bonds and release CO_2_. The high heat for decarboxylation can decompose phenolic compounds and terpenes. It may even lead to new compounds, such as highly reactive aldehydes and esters [12,13].

There is, nevertheless, a gap in the literature concerning the cannabis extraction processes guaranteeing a reproducible sample matrix [14,15]. Valizadehderakhshan et al. [16] compared different extraction methods for seed and trichomes in *Cannabis sativa* L. and reviewed various parameters that affect cannabinoid transformation after extraction.

This work reports a process set out to make the best Full-spectrum Extract Cannabis Oil for European and Global markets, starting with the premise that major and minor cannabinoids and terpene profile markers would be preserved throughout the process. The task was to maintain all the medicinal qualities of the original plant chemovars [17,18] and take away undesirables with no defined therapeutic value, like lipids, carbohydrates, chlorophyll, and polyphenols, in the gentlest possible way. The result is a better-tasting purified SOMAÍ full-spectrum extract of medicinal chemotype I: high-THC content with low-cannabidiol (CBD) content [19,20,21,22,23] *C. sativa* L. preserving genuine therapeutic properties, like the initial crude extract, while eliminating undesirable factors compromising adherence to therapy through improved organoleptic characteristics.

## 2. Materials and Methods

### 2.1. General Experimental Procedures

Chemical reagents, solvents, and cannabinoid reference standards for cannabinoid quantification were purchased from Sigma-Aldrich, Merck, or Carlo Erba and were used without further purification unless stated otherwise. The reagents used included ultrapure water (Milli-Q) supplied by the Milli-Q IQ 7000 Merck system, ethanol 96% (V3A033063B), acetonitrile (HPLC grade, P3C653153C), and methanol (HPLC grade, V2N046222N), all provided by Carlo Erba. Phosphoric acid (L2440) was sourced from Honeywell (Honeywell Fluka, HPLC, 85–90%, acc. Ph. Eur., BP, NF, Seelze, Germany). The reference standards used in the analytical methods were Cannabidivarin (CBDV, 2-H444030ME), Cannabidiolic acid (CBDA, 2-H495778AL), Cannabigerolic acid (CBGA, 2-H444251AL), Cannabigerol (CBG, 2-H497325ME), Tetrahydrocannabivarinic acid (THCVA, 2-H497522AL), Cannabichromene (CBC, 2-H542472AL), Cannabicyclolic acid (CBLA, 2-H441929AL), and Cannabichromenic acid (CBCA, 2-H497522AL), all provided by DR EHRENSTORFER. Additionally, Cannabinol (CBN, A0177171), Δ^9^-Tetrahydrocannabinol (Δ^9^-THC, A0180155), and Δ^8^-Tetrahydrocannabinol (Δ^8^-THC, A0180815) were supplied by RESTEK.

### 2.2. Plant Material

High-THC *Cannabis sativa* L. flower was acquired by SOMAÍ Pharmaceuticals (Carregado, Portugal). The samples for the present study consisted of eight brown and green clustered flowers from suppliers in Portugal, each accompanied by a certificate of identity. Reference specimens were preserved in the Plantoteca of both the supplier and SOMAÍ Pharmaceuticals. The flowers complied with the specifications outlined in [21]. All plant material used in this study was propagated by cuttings, cultivated, and harvested in mainland Portugal.

High-CBD *Cannabis sativa* L. was acquired as a crude extract, meeting the specifications defined in German Pharmacopeia DAB 2020 cannabis extract, standardized.

### 2.3. High-THC Cannabis sativa L. Flower Extraction

The starting material consisted of cannabis flower, THC-dominant, Ph. Eur. monograph-compliant, with a total THC content >14%. A total of 80 kg of cannabis flower, subdivided into 8 consecutive runs of 10 kg, was subjected to extraction and decarboxylation using the DEVEX CryoEXS100 equipment (DEVEX Verfahrenstechnik GmbH, Warendorf, Germany). The ethanol content was 96%, previously chilled at −40 °C. The extraction vessel was designed with a “double-jacked” and, as such, also cooled at −40 °C.

After extraction, with “percolation and residence contact time controlled, the tincture obtained (oleoresin dissolved in ethanol) is stripped from the biomass through the vacuum. A filter is installed in the circuit to ensure the removal of solid particles in suspension. “Overheated steam” is used in the final step for the complete removal of the remaining tincture from the biomass. The ethanol evaporation, a continuous thin-layer distillation process (contact surface at 100 °C), is followed by a decarboxylation reaction, in the same equipment, by increasing the temperature of the contact surface to 120 °C.

The decarboxylation reaction was monitored by thermogravimetric analysis using a Sartorius MA160 moisture analyzer (SARTORIUS AG, Göttingen, Germany), operated in real-time mode. The sample mass was continuously recorded under controlled heating conditions, and stabilization of the weight over time was used as an indirect indicator of the completion of decarboxylation. This approach assumes that the decarboxylation of acidic cannabinoids (e.g., THCA to THC) is associated with the release of CO_2_, leading to a measurable mass loss. The reaction was considered complete when no further significant mass loss (<0.001 g/min) was observed. At the end of this process, an average of 12.5 kg of FECO was obtained with a total THC concentration of >70%. The subsequent industrial experience confirmed a Drug Extraction Ratio (DER) of 5.5–9.5:1.

### 2.4. Purification Process

The purification process after extraction was achieved through a “short pass step” under a high vacuum, with VTA VK100 equipment (VTA Verfahrenstechnische Anlagen GmbH & Co. KG, Niederwinkling, Germany) using a thin-film (wiped-film) temperature exchanger and a concentric condenser.

The extract is previously heated, for melting purposes, using a vacuum oven at 70 °C. The crude oil extract (FECO) is submitted to a “previous passage” through the system to remove the more volatile components. At this stage, a pressure between 0.2 and 0.8 mbar is necessary, and the wipe-film surface is at a temperature of 110 °C (105–115 °C). The volatile components are trapped in one of the two cold traps set at −25 °C. This “previous passage” step is necessary to allow the purpose of purification at a high vacuum. The extract is then subjected to the purification process, with the pressure reduced to 0.15 to 0.35 mbar and a wiped film surface maintained at a temperature of 160 °C (155–165 °C). The flow rate is controlled to ensure standardized experimental conditions. From this process, distillate and residue are obtained.

A Sartorius MA160 moisture analyzer, although being a nonspecific analytical method, is used as In-Process Control (IPC), on the shop floor, to monitor the end of the decarboxylation process, for it is quick and easy to use. A qualification process has been previously performed to make the parallel between the HPLC quantification of the THCA/THC and the established analytical limit for the moisture analyzer results (not more than 2.5% at 120 °C for a 2 g sample for max 30 min).

The sample mass is continuously recorded under controlled heating conditions, and stabilization of the weight over time is used as an indirect indicator of the completion of decarboxylation. This approach is based on the assumption that the decarboxylation of acidic cannabinoids (e.g., THCA to THC) is associated with the release of CO_2_, leading to a measurable mass loss.

The reaction was considered complete when no further significant mass loss (<0.001 g/min) was observed, but the final quantification and the results reported are obtained in the final quality control analysis by HPLC, as further described. The purification process is not an “absolute” process but is related to a concentration ratio and rate, temperature, and pressure. As such, the residue obtained is consecutively submitted to further distillation processes, and the distillate fractions are collected and blended, up to a convenient yield and concentration in cannabinoids. Removal of unwanted compounds (e.g., fats, waxes, chlorophyll, volatiles) is achieved to a target purity for compliance with the formulation [24]. At the end of this process, an average of 88% of the initial crude (SOMAÍ FECO) weight is obtained as “Purified Full-spectrum Extract Cannabinoid Oil” (Purified SOMAÍ FECO). The total THC concentration is 80% on average. The final process (extraction + purification) presented a Drug Extraction Ratio (DER) of 7–12:1.

### 2.5. HPLC Analysis of Cannabinoids

All analytical procedures were performed by adapting the DAB methods [22] as described by Rosa (2023) [25].

#### 2.5.1. High-THC *Cannabis sativa* Flowers

To 500 mg of dry, ground cannabis flower samples in a 50 mL Falcon^®^ tube (Corning Falcon via VWR Portugal) with capacity up to 16,000× *g* ideal for freezing up to −80 °C protected from light, 20 mL of ethanol (96%) was added. The mixture was mixed at 300 rpm for 15 min and centrifuged at 3500 rpm for 5 min. The clear supernatant was transferred to a volumetric flask, and the extraction process was repeated twice as much with 12.5 mL of ethanol (96%) each time. Combined extracts were filtered through a 0.45 μm membrane filter, and the volume was adjusted to 50 mL with ethanol (96%). Two independent solutions were prepared for analysis.

#### 2.5.2. High-CBD and High-THC *Cannabis sativa* Extracts Sample Preparation

To analyze crude and purified extracts, 20 mg of product was weighed into a 100 mL volumetric flask and diluted to volume with ethanol (96%). The solution was vortexed until completely dissolved and filtered through a 0.22 μm membrane filter.

#### 2.5.3. Cannabinoid Analysis by HPLC-PDA

High-performance liquid chromatography (HPLC) equipment with a Photodiode Array Detector (PDA) included a Waters 2998 PDA Detector, a Quaternary Pump, a Quaternary Solvent Manager R, and an Autosampler (Waters Sample Manager FTN-R), all managed with Waters Empower software (version 3.7.0), all supplied by supplied by Izasa Scientific (Lisbon, Portugal), the official representative of Waters Corporation. Chromatography was performed using a pre-column (5 mm × 3 mm, octadecylsilylized silica gel, 2.7 μm) and an analytical column (150 mm × 3 mm, octadecylsilylized silica gel, 2.7 μm) at a flow rate of 1.0 mL min^−1^. The mobile phase consisted of solvent A (aqueous phosphoric acid, 8.64 g/L) and solvent B (acetonitrile), with an initial gradient elution of 64% B. Over 16 min, the gradient gradually increased to 82% B. The proportion of solvent B was then reduced to 64% for 1 min and held constant for 3 min to allow for re-equilibration. The column temperature was maintained at 40 °C, and the autosampler was set to 5 °C. The injection volume was 10 μL for both samples and standards. UV detection was performed at 225 nm for non-acid cannabinoids and 306 nm for acid cannabinoids.

For the quantification of Δ^9^-THCA, a certified reference standard of Δ^9^-tetrahydrocannabinolic acid (Δ^9^-THCA) was used. This standard was obtained from DR EHRENSTORFER, with purity ≥98% verified by the supplier’s Certificate of Analysis.

#### 2.5.4. Calibration Curves

Standard stock solutions were prepared with Δ^9^-THC and CBD (500 μg/mL, stock solution A), Δ^9^-THCA and CBDA (cannabidiolic acid) (500 μg/mL, stock solution B), and CBN (cannabinol) (20 μg/mL, stock solution C). For flower samples, the system standard solution (SSS) was prepared by combining stock solutions A, B, and C to achieve final concentrations of Δ^9^-THC and CBD at 10 μg/mL, Δ^9^-THCA and CBDA at 50 μg/mL, and CBN at 1 μg/mL. For extracts, calibration standards included Δ^9^-THC, CBD, and CBN at concentrations of 200 μg/mL, 200 μg/mL, and 2 μg/mL, respectively, as well as Δ^9^-THCA and CBDA at 20 μg/mL. Cross-verification standard solutions were independently prepared to validate calibration robustness. Calibration curves demonstrated linearity (R^2^ ≥ 0.99), and the relative standard deviation (RSD) of the response factors was ≤5%. Calibration was performed using six serial dilutions of the standard in methanol, covering the expected concentration range of sample extracts. The calibration curve was constructed by linear regression using the least squares method with no intercept (forced through the origin). The correlation coefficient (r^2^) was ≥0.99 for all analytes, and the %RSD for the response factor was ≤10%, fulfilling the defined acceptance criteria.

The same mathematical approach was applied to all samples, with specific adjustments based on sample type, particularly with respect to the moisture content in dried flowers and the reporting basis (dry weight vs. mass/mass). The concentration of the sample $C_{S\;}$, in μg/mL, was determined according to (1), where $W_{S}\;$corresponds to the mass of the sample, in mg, and ${\mathrm{Dil}}_{S}$, corresponds to the dilution factor of the sample in mL.(1)$$C_{S\;}=\frac{W_{S}}{{\mathrm{Dil}}_{S}}\;\times \;1000$$

The concentration of each cannabinoid ($C_{C}$) in the sample was then determined using the linear calibration curve generated for each analyte, and calculated from the chromatographic peak area ($A_{S}$) and calibration curve slope (m) using Equation (2).(2)$$C_{C}=\frac{A_{S}}{m}$$

For the dried flower samples, the cannabinoid concentration was further corrected to a dry weight basis, $C_{\mathrm{DB}}$ in %, with Equation (3) where $C_{C}$ is the concentration of the component in the sample determined through the calibration curve, in μg/mL in Equation (2); $C_{S\;}$ is the concentration of the sample, in μg/mL; and where LoD is the loss on drying percentage.(3)$$C_{\mathrm{DB}}=\frac{C_{C}\;\times \;100}{C_{S\;}\times \;\left(100-\mathrm{LoD}\right)}\;\times \;100$$

For extracts and purified extracts, cannabinoid content was expressed as a percentage by weight (% *w*/*w*) using Equation (4), where $A_{S}$ is the sample response (area), the $C_{\mathrm{ST}}$ is the standard concentration (μg/mL) and $A_{\mathrm{ST}}\;$is the average response (area) of the standard solution injections.(4)$$C_{\left(w/w\right)}=\frac{A_{S}\times C_{\mathrm{ST}}}{A_{\mathrm{ST}}\times C_{S}}\;\times \;100$$

In both matrices, the total concentrations of Δ^9^-tetrahydrocannabinol (total THC) and cannabidiol (total CBD) were calculated by accounting for the presence of their respective acidic precursors, Δ^9^-THCA and CBDA. This was performed using a molecular decarboxylation correction factor of 0.877, which adjusts for the mass lost as carbon dioxide during decarboxylation. The equations used were as follows:(5)$$Total\;THC=THC\;\left(\%\right)+0.877\times THCA\;\left(\%\right)$$(6)$$Total\;CBD\;\left(\%\right)=CBD\left(\%\right)+0.877\times CBDA\;\left(\%\right)$$

Retention times (RTs) and relative retention times (RRTs) for cannabinoids used in these calculations are summarized in Table 1, providing essential reference points for chromatographic identification and quantification.

System suitability was confirmed before sample analysis by verifying the retention time stability, peak efficiency (≥2000), symmetry (0.8–1.5), and resolution between Δ^9^-THC and Δ^8^-THC (≥1.2 for flower, ≥1.5 for extracts). Cross-verification solutions, prepared independently of the standard curve, showed recovery values within 85–115% for Δ^9^-THCA and other cannabinoids.

### 2.6. Total Phenolic Content

Total phenolic content was measured using the Folin–Ciocalteu method, as described by Cicco et al. (2009) [26], Kupina et al. (2018) [27], and Waterhouse (2003) [28]. A calibration curve was established using five gallic acid (Sigma-Aldrich, represented in Portugal by Merck Portugal–Millipore Sigma, Algés, Portugal) solutions with concentrations between 50 and 500 mg/L. Briefly, 100 µL of Folin–Ciocalteu reagent (Sigma-Aldrich) was mixed with 1.58 mL of distilled water and added to 20 µL of extract solution in methanol (10 mg/mL, MeOH, Carlo Erba, from Proquinorte S.A. via Bioportugal, Lda, Porto, Portugal). The mixture was incubated at room temperature for 7 min, after which 300 µL of a 20% sodium carbonate solution (Na_2_CO_3_, Sigma-Aldrich from Merck Portugal–Millipore Sigma) was introduced. The reaction mixture was then incubated at 40 °C for 30 min to allow color development. Absorbance was recorded at 750 nm using a UV/VIS spectrophotometer Cary 60 (Agilent Technologies Santa Clara, CA, USA), supplied in Portugal by Izasa Scientific, the official distributor.

### 2.7. Total Flavonoid Content

Total flavonoid content was determined based on the method outlined by Navarro J.M. et al. [29] and Shraim et al. [30]. A calibration curve was prepared using five quercetin (Sigma-Aldrich via Merck Portugal–Millipore Sigma) solutions with concentrations between 5 and 250 mg/L. The procedure began by mixing 37.5 μL of 5% sodium nitrite (NaNO_2_) with 125 μL of the extract (10 mg/mL) and incubating for 6 min. Following this, 75 μL of 10% aluminum chloride (AlCl_3_) was added and allowed to incubate for 5 min. The reaction was finalized by adding 250 μL of 1 M sodium hydroxide, with the final volume adjusted to 1.25 mL with distilled water. Absorbance was measured at 510 nm using a microplate reader, and all experiments were performed in triplicate.

### 2.8. Total Chlorophyll Content

For the total chlorophyll content, the method of Arnon et al. (1949) [31] and Tiago et al. (2022) [32] was adapted. Approximately 0.1 g of extract was diluted in 10 mL of 96% ethanol (*v*/*v*). After centrifugation (6000 rpm for 15 min), the absorbance of the supernatants was measured at 663 nm and 645 nm. The contents of chlorophyll a (CHL_a_), chlorophyll b (CHL_b_), and total chlorophyll (CHL_Total_) were calculated using Formulas (7)–(9).(7)$${\mathrm{CHL}}_{a}=(12.25\;\times \;{\mathrm{Abs}}_{663\mathrm{nm}\;})\;-\;(2.79\;\times \;{\mathrm{Abs}}_{645\mathrm{nm}\;})$$(8)$${\mathrm{CHL}}_{b}=(21.50\;\times \;{\mathrm{Abs}}_{645\mathrm{nm}\;})\;-\;(5.10\;\times \;{\mathrm{Abs}}_{663\mathrm{nm}\;})$$(9)$${\mathrm{CHL}}_{\mathrm{total}}\;={\mathrm{CHL}}_{a}\;+{\mathrm{CHL}}_{b}$$

### 2.9. Waxes Determination

Wax quantification was performed through a winterization process, where 100 mg of the sample was dissolved in 10 mL of 96% ethanol and frozen for 24 h to precipitate the waxes. The mixture was centrifuged at 6000 rpm for 15 min, and the resulting pellet was left to dry overnight. The wax quantification was performed in triplicate and calculated as follows (10):(10)$${\mathrm{Waxes}}_{\mathrm{total}}\;\left(\%\right)\;=\frac{\mathrm{Extracted}\;\mathrm{waxes}\;\left(g\right)}{\mathrm{Feed}\;\mathrm{mass}\;\left(g\right)}\;\times \;100$$

### 2.10. Terpene Quantification by GC-FID

Terpenes were quantified using gas chromatography with flame ionization detection (GC-FID) on an Agilent HP-5 column (30 m × 0.32 mm, 0.25 µm). Sample preparation involved extraction with ethanol 96%, with or without internal standard (IS) solution (n-nonane, 100 or 1000 µg/mL), followed by filtration through a 0.22 µm membrane. Specific dilutions (100, 2.5, or 0.125 mg/mL) were prepared depending on sample type (flower, resin, extract, excipient). Chromatographic conditions included split injection (1:20) at 250 °C, using helium as carrier gas (1 mL/min, constant flow). The oven program was as follows: 65 °C (2 min), ramp to 160 °C at 7.5 °C/min, then to 240 °C at 25 °C/min (hold 3 min). The FID operated at 300 °C with H_2_ (30 mL/min), air (300 mL/min), and N_2_ (30 mL/min).

Calibration was performed using seven terpene standard levels (5–400 µg/mL) prepared from certified reference mixtures. A control point (100 µg/mL) was injected throughout the sequence to ensure analytical consistency. Peak identification was based on the relative retention time (RRT) to the IS, with a tolerance of ±0.02. The method was validated over a range of 5–200 µg/mL, with LOD of 2 µg/mL and LOQ of 5 µg/mL. Quantification was based on internal standard normalization and external calibration curves, with results expressed in mg/g and relative% composition.

### 2.11. Sensory Analysis

Two sensory tests were performed. Test 1 was carried out between the 10THC:10CBD crude edible FECO sample solution and 10THC:10CBD edible purified FECO sample solution, both in MCT oil, designated as A and B, respectively. Test 2 compared the same purified FECO-based SOMAÍ 10 THC:10 CBD edible sample solution and a more similar solution in MCT oil already granted ACM (Marketing Authorization by the National Authority) as herbal preparation to the market, 5THC:20CBD, labeled as samples C and D, respectively. The sensory analysis was carried out by a panel of 12 trained assessors, following the criteria outlined in ISO Standard 8586-1 (1993) [33]. All participants were members of the sensory evaluation panel at INIAV, each with over 200 h of experience in food analysis. The panel consisted of 81.8% women and 18.2% men, with ages ranging from 18 to 70 years.

Before the evaluation, the assessors were informed about the nature of the substances and the administered doses. The need to not swallow the sample was reinforced, as well as the analysis procedures and the use of collected data, which was strictly for research purposes. Anonymity and compliance were also assured in line with the General Data Protection Regulation (GDPR).

For sample preparation and presentation, each sample was provided in individual, transparent, coded dosing syringes containing a sub-therapeutic dose to minimize potential physiological effects and ensure the safety of the evaluators. To prevent identification bias, the samples were randomly coded. Each assessor was asked to evaluate two samples using a questionnaire divided into three sections. Firstly, for the hedonic analysis, the assessors rated attributes like appearance, smell, color, taste, and overall appreciation on a 7-point scale, where 1 meant “I disliked it very much” and 7 meant “I liked it very much”. Then, for the intensity profile, the assessors rated the intensity of the attributes (color, smell, and taste) on a 7-point scale, where 1 indicated “not intense at all” and 7 indicated “very intense”. Lastly, the assessors answered open-ended questions, including the following: “Is the taste of the sample pleasant?” “Can you identify any specific tastes?” (e.g., herbal, fruity, citrus, or others), “Is there any immediate discomfort or rejection caused by the taste?” and “How long does the taste persist, and how does it influence your decision to continue with potential daily use of the oil?”.

Between samples, the assessors were asked to drink water and eat a biscuit to neutralize any lingering perceptions. The samples evaluated followed the same coding and presentation protocol as described in Test 1. By following this standardized methodology, the robustness of the results was ensured, minimizing any biases and promoting the reliability of the sensory analysis of the cannabis oils.

The evaluations took place in standardized individual booths under controlled lighting conditions, with the temperature maintained at 21 ± 2 °C.

### 2.12. Statistical Analysis

The results from the hedonic analysis and intensity profile were analyzed using STATISTICA 13.3 software (StatSoft, Kraków, Poland). The data are presented as mean values (M) ± standard deviation (SD). The sensory analysis data were subjected to the Tukey test with Honest Significant Difference (HSD) (*p* < 0.05).

## 3. Results

### 3.1. Cannabinoid Analysis

To identify the cannabinoids, reference substances were acquired and individually injected with a concentration of 2 μg/mL to compare and confirm their retention times, according to the DAB monograph [22]. The overall composition of cannabinoids quantified by HPLC-PDA in the herbal medicinal substance, the dried flowers, and the herbal medicinal crude and purified extracts (SOMAÍ FECO and purified SOMAI FECO, respectively) is presented in Table 2.

### 3.2. Non-Cannabinoid Analysis

Compounds such as phenolic compounds, flavonoid compounds, waxes, and chlorophyll contribute to the sensory properties of the extracts, potentially affecting the aroma, taste, and overall perception. Therefore, the non-cannabinoid composition of the flower (herbal medicinal substance), crude extract (SOMAÍ FECO), and purified soft extract (purified SOMAÍ FECO) was analyzed to evaluate the impact of extraction and purification (Figure 1 and Appendix A).

### 3.3. Terpene Analysis

To gain further insight into the composition of terpenes, additional analysis was conducted to identify and quantify individual terpenes in the flower, crude extract, and purified extract (Figure 2). Of the 30 terpene standards analyzed, 26 were quantified across the different process stages. To complement the analysis, two current herbal preparations (OS1 and OS2) were compared. The results are summarized in Appendix A, providing a detailed breakdown and recovery by precise, consistent, and tailored terpene profiles, by controlled reintroduction of the characteristic terpenes across the processing stages of final herbal medicinal preparations as oral solutions from a premium line of SOMAÍ Origins and Senses, which were created under the highest EU-GMP quality standards.

### 3.4. Sensory Tests

Sensory tests were performed to understand the natural contribution of cannabinoid extractive FECO without the appealing terpene influence by comparing formulations of purified SOMAI FECO (sample A), purified SOMAI FECO-based solutions (samples B and C), and a product from the market (sample D). As some panel members were common and to avoid results bias, the same formulation with purified SOMAI FECO received two sample references (B and C).

In Figure 3, the average results and standard deviation (M ± SD) of the hedonic evaluation of attributes such as appearance, smell, color, taste, and overall appreciation are presented for samples A and B of sensory test 1. As observed, all attributes were rated between scores 4 (“Indifferent”) and 6 (“Liked”) for both samples.

Figure 4 shows the intensity evaluations of attributes such as color, smell, and taste for samples A and B, respectively. Appendix A presents the open-ended questions related to the perception and acceptance of the SOMAÍ crude and purified herbal high-THC preparations.

The same attributes evaluated in Test 1 were analyzed for samples C and D in Test 2, using a panel with a different demographic profile, as previously mentioned. This panel was smaller and primarily composed of older participants. Figure 5 presents the mean values (M) and standard deviation (SD) of the intensity evaluations for attributes such as color, smell, and taste in samples C and D.

The results in Figure 6 consist of the means and standard deviation (M ± SD) of the intensity evaluations of attributes such as color, smell, and taste for samples C and D. The open-ended questions related to the perception and acceptance of the solutions are presented in Appendix A.

## 4. Discussion

### 4.1. Sensory Test Analysis

The results from the two sensory tests (Test 1 and Test 2) provide relevant insights into the sensory attributes of cannabis solutions and offer guidance for product development.

In Test 1, the demographic composition of the evaluation panel played a crucial role in sensory perception. The panel included participants aged 18 to 70, ensuring moderate sensory response diversity. However, the predominance of younger participants (18–30 years, 54.5%) may have influenced the results, as younger consumers tend to prefer pronounced tastes and aromatic profiles. Participants aged 31–50 years (36.4%) balanced the sample, whereas older participants (51–70 years) were underrepresented (9.1%), limiting conclusions about their preferences. Regarding gender distribution, the panel had a female majority (72.7% female vs. 27.3% male). Since women generally exhibit heightened olfactory and gustatory sensitivity, this may have contributed to a greater ability to identify subtle sensory characteristics. The demographic characteristics of the panel highlight the importance of achieving a more balanced and inclusive sample to ensure broader consumer representation.

The hedonic and intensity evaluations in Test 1 (Figure 7) demonstrated that all attributes, including appearance, smell, color, taste, and overall appreciation, were rated between 4 (“Indifferent”) and 6 (“Liked”) for both the SOMAÍ FECO and purified FECO samples (A and B). Sample B (purified SOMAÍ FECO) received slightly higher overall acceptance due to its more consistent sensory profile, while sample A (SOMAÍ FECO) showed greater variability in taste, suggesting more divergent opinions among evaluators. Significant differences were observed in color and aroma intensity between the two samples. Although both were well accepted, sample B demonstrated greater uniformity in acceptance, whereas sample A’s stronger taste profile led to a more polarized reception. The comparison of these samples indicated that sample B had higher overall acceptance, with most panelists rating it favorably. In contrast, despite being described as pleasant, sample A exhibited stronger taste persistence, which could be a rejection factor for some consumers. Both samples were predominantly described as herbal, although sample A also had a slight fruity note.

In Test 2 (Figure 8), the same sensory attributes were evaluated in samples C, the same purified SOMAÍ FECO identical to sample B, (20 mg/mL major cannabinoids) in MCT oil, and D, a product from the market (25 mg/mL major cannabinoids), but with a panel composed of participants from a higher age range and fewer overall participants. While sample C was rated higher across all attributes, these differences were not statistically significant. However, a notable variation was observed in color perception (*p* < 0.5), highlighting a visual preference for sample C.

The results showed that sample C had stronger acceptance, particularly in appearance and color, with five panelists finding it highly acceptable, whereas sample D received positive evaluations from only two panelists.

In terms of taste, sample C (identical to sample B) maintained the herbal profile identified in Test 1, while sample D introduced citrus notes. Although these notes were considered interesting, they did not overcome the perception of imbalance, leading to lower acceptance. Both samples had persistent aftertastes, but sample D was perceived more negatively, with some panelists finding it unpleasant.

The sensory analysis of cannabis solutions in MCT oil revealed notable differences between the evaluated samples, highlighting sensory attributes that directly impact product acceptance. In Test 1, purified SOMAÍ FECO as sample B was slightly more accepted due to its consistency and lower variability, while the intensity and persistence of SOMAÍ FECO as sample A led to more polarized responses, indicating the need for sensory adjustments to appeal to a wider audience.

In Test 2, sample C (identical to B) received higher acceptance, particularly for visual attributes such as appearance and color. Sample D’s citrus notes were intriguing but did not enhance the overall acceptance of this full extract preparation in the market, as its imbalance and after-taste were perceived negatively. Although all samples were generally well-received, taste persistence and aromatic intensity should be carefully considered, particularly for sensitive consumers.

### 4.2. Extraction and Purification Process

Ethanol extraction is a well-established method for cannabis flower processing. The extraction parameters, such as ethanol concentration (the solvent), temperature, and contact time, have been optimized to guarantee reproducible extraction conditions and a solvent with adequate polarity. Ethanol recovered from the distillation process can be re-utilized in the process, within the concentration limits established, for economic and environmental reasons.

A key step in the process is decarboxylation, which converts acidic cannabinoids, like THCA, into their non-acidic, therapeutically active forms, such as THC. This reaction is performed by increasing the temperature up to less than 120 °C under atmospheric pressure conditions, keeping a thin layer in the heating transfer. This allows optimization of the process, avoiding foam generation and unnecessary stress exposure on the extract. The end of the reaction is monitored through sampling along the process and thermos-scale loss on the weight index.

The purification process is achieved through a “short pass distillation process”, a well-known technique performed under a deep vacuum and with a wiped-film system. Since the crude oil is already decarboxylated before using the short-pass distillation process, the usage of the machine is for purification not distillation as traditionally required.

Hence, to further purify the extract, a state-of-the-art wiped-film system was employed under a deep vacuum (0.2 mbar or lower) to precisely separate compounds based on their boiling points while preserving cannabinoids. Since all decarboxylated cannabinoids have similar boiling points ranging between 105 and 180 °C, maintaining a vacuum depth of 0.2 mbar or lower keeps distillation temperatures below 175 °C with an incredibly short residence time, therefore minimizing thermal degradation. The process maintains the product at 110–115 °C while the heat transfer surface reaches less than 165 °C, optimizing the separation efficiency. SOMAÍ carries out purification processes for high-THC and high-CBD extracts, as both can be part of the oral solutions it offers to the market. However, while the THC process starts from the cannabis flower, CBD purification is performed on a previously acquired crude extract (Figure 9). Therefore, this study focuses on detailing the purification pathway for THC-purified soft extract [24].

In this system, the extract is rapidly heated upon entering a central “wiper” area that moves clockwise, effectively directing the vapor downwards. At the same time, it naturally rises towards the condenser. This motion creates theoretical plates, or potential condensation zones, effectively eliminating unwanted co-distilled compounds directed to the residue waste. The primary condenser is set to 65–75 °C, selectively condensing and retaining the target cannabinoids to further guarantee purity. These tight controls and deep vacuum conditions facilitate the cannabinoids within the full-spectrum oil to co-distill and produce the SOMAÍ Full-spectrum Extract Cannabinoid Oil corresponding to the herbal medicinal preparation.

### 4.3. Cannabinoid Content in Herbal Extracts

The purified SOMAÍ FECO (Table 2) is subjected to quality control evaluation to confirm a total cannabinoid content in the extract of over 90% while maintaining the same ratio between them as their presence in the plant, considering the expected transformation of acidic in non-acidic forms. This process is designed to retain medicinal compounds while eliminating undesirable components, such as bitter-tasting and non-defined therapeutic substances. The observed total cannabinoid content in the SOMAÍ’s *C. sativa* flower of 21.9 ± 2.2% aligns with, and in some cases exceeds, the range of 15.77–20.37% reported in the literature [34] for modern drug-type chemovars, further confirming the potency of the samples analyzed.

### 4.4. Deep-Cooled Ethanol Extraction

The SOMAÍ extraction process begins with terpene extraction and then deep-cooled ethanol (−30 °C to −60 °C). This approach is aimed at a more selective solubilization of target compounds, based on ethanol’s dual polarity, since it exhibits polar (hydroxyl group) and nonpolar (ethyl chain) attributes. At subzero temperatures, it behaves primarily as a nonpolar solvent, significantly reducing the solubility of polar compounds, such as chlorophyll, compared to it in ambient-temperature ethanol. This is particularly important since the extraction method is specific for extracting compounds from the resin glands.

More precisely, the process consists of a gentle wash of the cannabis flower with a time-controlled contact period and solvent percolation. While the flower remains stationary, the ethanol washes over the surfaces where trichomes are present. The polarity attributes of ethanol allow it to gently rupture the trichome walls without additional energy, even at cold temperatures. The FECO is obtained by distilling ethanol from the oleoresin tincture previously obtained.

In the purification step (short-path distillation), the vapor generated under a deep vacuum that reaches the primary fraction condenser will only condense if it contains cannabinoids. Volatile organic compounds (VOCs) that may have co-distilled will travel beyond the condenser and be collected at a secondary cold trap set to −90 °C. This results in a purer FECO that contains all the cannabinoids that are present in the plant. The purification effectively separates the cannabinoids from polyphenols and lipids. Residual solvents in the oil, such as ethanol, will immediately boil and be collected within cold traps at a temperature of −90 °C. Lipids, chlorophyll, and polyphenols will not be distilled and instead will travel to the waste of the process. The separation is nearly perfect with very small amounts of undesirables that will co-distill, but a vast majority of the undesirable, foul-tasting compounds will be left in the residue waste of the purification because their boiling points are either much lower or higher than the cannabinoids.

The final purified FECO oil produced will have a total active cannabinoid concentration of >95%, with the remaining <5% being a combination of compounds that we have yet to identify, waxes, and traces of chlorophyll. The synergistic effects of the cannabinoids will remain the same because the exact ratio of cannabinoids in the plant to the extract is retained; there are no additions or subtractions of cannabinoids.

The preservation of acidic cannabinoids requires extraction at room temperature [35]. To decarboxylate acidic cannabinoids into neutral form, high temperatures are needed during extraction, although a higher temperature may result in the loss of some terpenes and minor constituents [36]. Therefore, selecting an appropriate extraction procedure will benefit future stages of development by minimizing the requirements for refinements [15].

### 4.5. Non-Cannabinoid Content

Throughout cannabis processing, non-cannabinoid compounds change in different proportions, especially during the transition from flower to SOMAÍ FECO. As illustrated in Figure 1, phenolics and flavonoids increase approximately 10-fold and 8-fold, respectively, in SOMAÍ FECO compared to flower, while chlorophyll and waxes are almost entirely removed. In purified SOMAÍ FECO, these ratios remain stable, with flavonoids showing a slight increase (9-fold) relative to flower (see corresponding values in Table 3). Evaluating these compounds is crucial for understanding the chemical transformations occurring during extraction and their effects on the final product. This analysis provides insight into which compounds are preserved, lost, and/or recovered during the overall stages of the process.

It is intended to become clear that waxes and chlorophyll reduce very significantly, while no changes were observed in total phenolic content.

Furthermore, our study reported the phenolic content in the flower to be 0.14 ± 1% (*w*/*w*), thus being within values found in previous studies [37,38] that reported values of 0.19% (*w*/*w*) (1.938 ± 0.01 mg GAE/g ext) on ethanolic extracts and 0.08% *w*/*w* (0.823 mg GAE/g ext). In contrast, Drinić et al. (2018) [39] observed a notably higher phenolic content after extraction with 90% ethanol, ranging from 0.585% to 0.643% (*w*/*w*) (5.85–6.43 mg GAE/g dw) for mature and young hemp, respectively, while another study reported values as low as 0.03% (*w*/*w*) (0.26 ± 0.02 mg GAE/g ext) for a mixture of flower and leaves of cannabis plant [32].

The flavonoid content in the flowers in this study was found to be 0.12 ± 0.1% (*w*/*w*), aligning with the findings of Jin et al. (2020), who reported values in cannabis flowers ranging from 0.07% to 0.14% [34]. Higher flavonoid content between 0.3 and 0.4% *w*/*w* (3.18 and 3.90 mg QE/g dw) was reported by Drinic et al. [39], whereas lower values of 0.03% in ethanolic extract (0.272 mg QE/g dw) were also reported by Ahidar et al. 2024 [38]. Tiago et al. (2022) [32] reported the highest values of 3.2% *w*/*w* in flavonoid content (32.25 ± 4.1 mg QE/g ext) for a mixture of leaves and flowers rather than flowers alone.

These findings are consistent with the trend that both flavonoid and phenolic content tend to vary significantly based on the cannabis chemovar and the age of the plant.

Chlorophyll content was significantly lower in SOMAÍ FECO (0.11 ± 0.04 × 10^−3^% *w*/*w*) and purified SOMAÍ FECO (0.03 ± 0.02 × 10^−3^% *w*/*w*) compared to the flower (3 ± 1% *w*/*w*), and substantially lower than the ones reported in the literature [32] of 0.9% before winterization and 0.3% after winterization. Notably, the extraction and purification process employed effectively removes the majority of chlorophyll content, reducing it to residual values of 0.03 ± 0.02 × 10^−3^% (*w*/*w*), representing a substantial reduction of ca. 10 000:4 in the purified FECO.

The wax content in the flower was measured at 9.2 ± 3.1% (*w*/*w*), which is in the same range as those reported by Tiago et al. (2022) [32], who found 7.75 ± 2.2% (*w*/*w*) in cannabis extracts. This consistency aligns with the well-documented co-extraction of waxes during cannabinoid and bioactive compound extraction. Furthermore, purification processes used by SOMAÍ (deep-cooled ethanol), which target plant impurities, including waxes and chlorophyll, are known to reduce chlorophyll levels. This is crucial for producing high-quality cannabis extracts, as chlorophyll can negatively impact color and taste and potentially pose health risks at high concentrations [40].

Additionally, considering the flower as the reference (100% compound content), the extraction process led to a concentration of phenolic and flavonoid compounds while reducing the relative concentration of impurities such as chlorophyll and waxes, as depicted in Figure 10. For instance, the phenolic content increased tenfold, reaching 10 times the original value (1000%) in both crude and purified SOMAÍ FECO. Similarly, flavonoids increased to 833.3% in SOMAÍ FECO and 916.7% in purified SOMAÍ FECO, while chlorophyll showed a significant reduction, dropping to 0.004% and 0.001% of the original value, respectively. A similar trend was observed for waxes, decreasing to 0.045% in SOMAÍ FECO and 0.041% in purified SOMAÍ FECO, while no changes were observed in phenolic content throughout the purification process.

### 4.6. Terpene Analysis

SOMAÍ developed a step of taking out the terpenes first and reintroducing them. Table 3 illustrates the original terpenes trapped and isolated during the extraction and purification stages [24] from one flower supplier. Therefore, subsequent reintroduction is an added benefit to recreate the herbal substance profile of natural “senses” while avoiding heat-related degradation and the potential production of undesirable byproducts.

On the other hand, to further understand the impact of the extraction and purification process on the overall terpene profiles, a subset of 14 of the 24 quantified terpenes listed in Appendix A were selected for analysis based on their presence in the flower at concentrations exceeding 0.01 % *w*/*w*. As terpenes play a key role in aroma and taste, understanding their changes in composition may help to manage the impact on the sensory profile of the extracts. Thus, their levels were examined across the flower, SOMAÍ FECO extract, and purified SOMAÍ FECO extract to evaluate the influence of extraction and purification when targeting the cannabinoids (Figure 2).

Despite being present in average quantities in the flower, alpha-bisabolol becomes a predominant terpene in the studied purified extract related to the herbal substance supplier, herein studied in depth. This is noteworthy given its potential calming and soothing effects as well as its significant bioactivities, including anti-inflammatory, anti-nociceptive, and anti-tumor effects, which may offer a challenge to research on enhancing both the therapeutic value and sensory appeal of the final product [41]. Beta-caryophyllene is the most abundant terpene in the cannabis flower variety herein studied and respective crude extract, and it is still present in the purified extract. Given its anti-inflammatory and potential therapeutic properties, as well as its unique ability to bind to the CB2 cannabinoid receptor in the mammalian endocannabinoid system, research is ongoing to find clinical evidence that reintroducing beta-caryophyllene into purified extracts seeks enhancement of therapeutic potential [42,43].

Highly volatile terpenes, such as alpha-pinene and camphene, are significantly reduced or may become undetectable during extraction and purification. This is consistent with the thermal stresses involved, which are known to contribute to the evaporation or degradation of these compounds. Also, several terpenes, such as D-limonene and myrcene, show substantial reductions between the flower and purified extract, emphasizing the selective retention of specific compounds through purification. Only the consistent and precise control of production could afford this knowledge, enabling full-spectrum extracts to be redesigned with nature-recovered senses to deliver complete “SOMAÍ senses” oral solutions.

The observed trends highlight the selective but highly conservative nature (79% still qualitatively present) of the nature terpene profile in extraction and purification processes, balancing between the reduction in impurities and retention of valuable non-cannabinoid compounds, critical for producing high-quality herbal extracts with both medicinal and sensory appeal.

Hence, developing a pharmaceutical process with chemically monitored natural intermediates at all stages is advantageous since it eliminates potential unknown variables. Additionally, as an added benefit, previously separated terpenes can be reintroduced to recreate the herbal substance profile or an infinite sense of variations while avoiding heat-related degradation and the potential production of undesirable byproducts. SOMAÍ’s approach can mitigate terpene loss during extraction and prevent terpene degradation during the decarboxylation of the crude extract.

As a proof of concept for this method, the composition of two premium oral solutions was included in addition to this FECO study from its proprietary SOMAÍ Origins and Senses lines, which was manufactured under the strict EU-GMP quality guidelines through the reintroduction of terpenes post-purification. The profile comparison of terpenes of these oral solutions and the flower as the starting material is depicted in Figure 11, demonstrating the reinstatement of nature-identical potential therapeutic and aromatic features. It should be noted that in the original solutions, which are cannabis herbal preparations, the purified FECO extracts are relatively diluted in MCT oil, and is the reason why they exhibit a proportionally lower intensity of all the constituents relative to the herbal substance.

This approach enables precise formulation by selectively incorporating terpenes and performing clinical studies to investigate any desired synergic or additive therapeutic outcomes for patients and healthcare providers. The obtained finished products should retain the full spectrum of therapeutic compounds while being free of secondary metabolites that contribute to undesirable taste profiles and decreased stability.

Many scientific publications suggest that these combinations of cannabinoids and terpenes can contribute to additive or super-additive effects compared to isolated cannabinoids [44]. André et al. [45] offer a detailed analysis of the entourage effect in medicinal cannabis products. The authors conducted a comprehensive review and noted that although there are interesting preclinical results, clinical evidence is still limited. They emphasize the need for more clinical studies to investigate the clinical relevance of the entourage effect and to determine the optimal combinations of cannabinoids and terpenes for different medical conditions. Therefore, while the specific therapeutic effect of terpenes is not clinically investigated in cannabis preparations, their profile as is in the herbal substance is desirable to guarantee a dose–effect outcome with reproducibility within a therapeutic plan with medicinal cannabis. Hence, based on this study, SOMAÍ has developed an advanced process to remove terpenes before purification. By doing so, these highly volatile compounds remain intact and can be reintroduced later, preventing their degradation. This innovative step can further enhance sensory experience and explore any therapeutic potential.

### 4.7. Organoleptic Properties of FECO Formulations

A comparative assessment of the organoleptic characteristics of different final products (produced from crude extract, purified crude extract, and a commercial market formulation) was conducted using a trained sensory panel. The results support the advantages of the purified FECO approach, demonstrating that purification improves the sensory acceptance of cannabis-derived formulations while maintaining the integrity of key bioactive compounds. These findings reinforce the necessity of a controlled extraction and purification strategy for producing high-quality, reproducible medicinal cannabis products with optimized sensory and therapeutic attributes.

## 5. Conclusions

This study highlights the effectiveness of SOMAÍ’s extraction and purification processes in optimizing the sensory and therapeutic quality of full-spectrum medicinal cannabis extracts for herbal preparations. The method employed was explicitly tailored to collect cannabinoid fractions from the natural cannabis flower oils. The resulting purified full-spectrum extract is selectively concentrated in the profiled beneficial compounds naturally found within the flower, successfully removing compounds lacking scientifically proven therapeutic value. While these unwanted compounds will have an impact on the final taste of the decarboxylated full-spectrum oil, herbal medicines created from unpurified full cannabis extract oil will, therefore, have a range of tastes, such as earthy, bitter, and grassy, all of which are strong and generally regarded as unpleasant for therapeutic adherence. The process developed herein is to obtain a standardized THC-dominant crude extract. The same process is applied to CBD-dominant crude extract. With these two extracts, the pharmaceutical company can prepare medicines with a precise and previously defined ratio between the two main bioactive components, an objective not achievable if depending exclusively on the present ratio in the herbal substance.

## Figures and Tables

**Figure 1 pharmaceutics-17-00848-f001:**
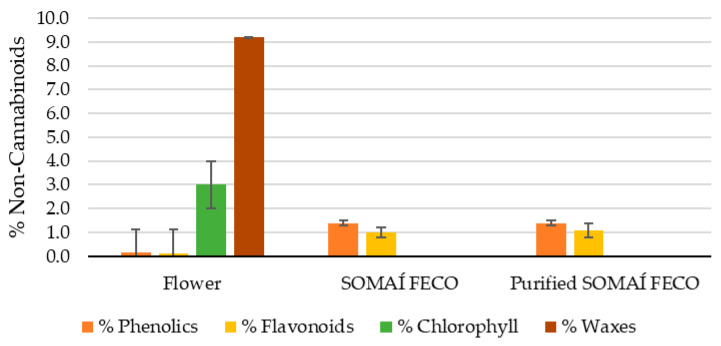
Content of phenolics, flavonoids, chlorophyll, and waxes (% *w*/*w*) in flower, SOMAÍ FECO, and purified SOMAÍ FECO.

**Figure 2 pharmaceutics-17-00848-f002:**
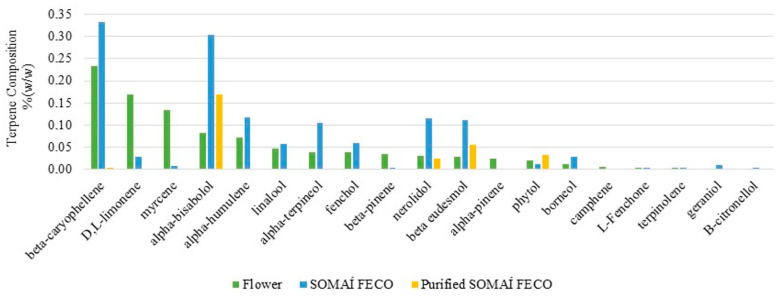
Terpene composition (% *w*/*w*) in flower, SOMAÍ FECO, and purified SOMAÍ FECO along the extraction and purification processes, well recovered for reintroduction in final solutions, see (Figure 11).

**Figure 3 pharmaceutics-17-00848-f003:**
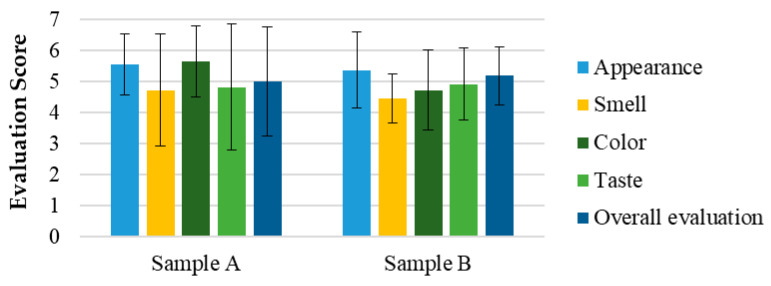
Evaluation of the sensorial attributes assessment of the SOMAÍ cannabis samples A (high-THC SOMAÍ FECO) and B (purified high-THC SOMAÍ FECO): appearance, smell, color, taste, and overall acceptability. Results are presented as means ± standard deviation (SD).

**Figure 4 pharmaceutics-17-00848-f004:**
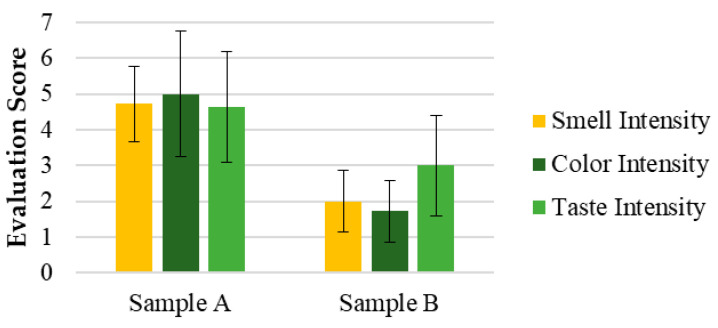
Evaluation of the intensity of the sensorial attributes of the SOMAÍ cannabis samples A (crude high-THC FECO) and B (purified high-THC FECO): color, smell, taste. Results are presented as means ± standard deviation (SD) to illustrate individual sample profiles.

**Figure 5 pharmaceutics-17-00848-f005:**
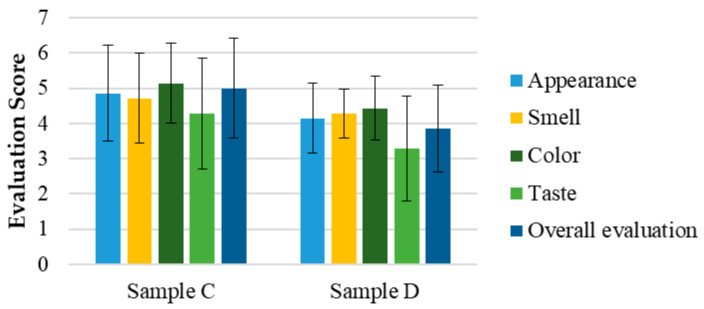
Evaluation of the sensorial attributes assessment of the final herbal preparations of cannabis samples (purified high-THC and/or high-CBD SOMAÍ FECO) as oral solutions in MCT oil: C and D (marketed oral solution) for appearance, smell, color, taste, and overall acceptability. Results are presented as means ± standard deviation (SD).

**Figure 6 pharmaceutics-17-00848-f006:**
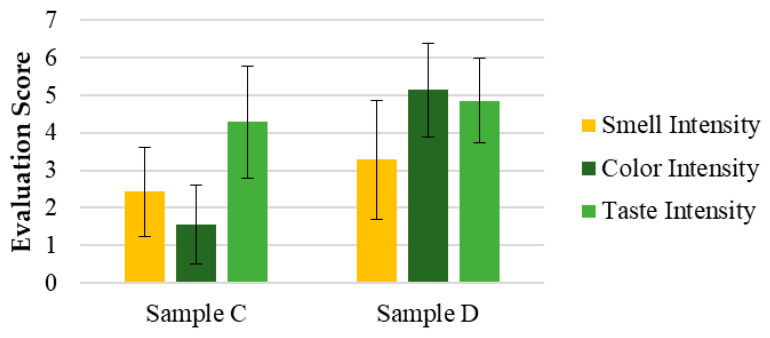
Evaluation of the intensity of the sensorial attributes of the cannabis samples: color, smell, taste. Results are presented as means ± standard deviation (SD) to illustrate individual sample profiles.

**Figure 7 pharmaceutics-17-00848-f007:**
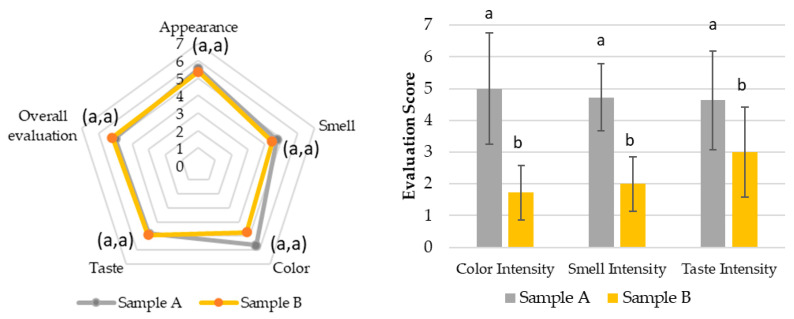
Sensory evaluation of FECO cannabis samples. (**Left**) Radar chart showing mean scores for appearance, smell, color, taste, and overall evaluation of samples A (SOMAÍ FECO) and B (purified SOMAÍ FECO). (**Right**) Bar graph comparing color, smell, and taste intensity between samples A and B. Values represent mean scores given by the panel, with statistically significant differences (*p* < 0.05) indicated by letters for each attribute.

**Figure 8 pharmaceutics-17-00848-f008:**
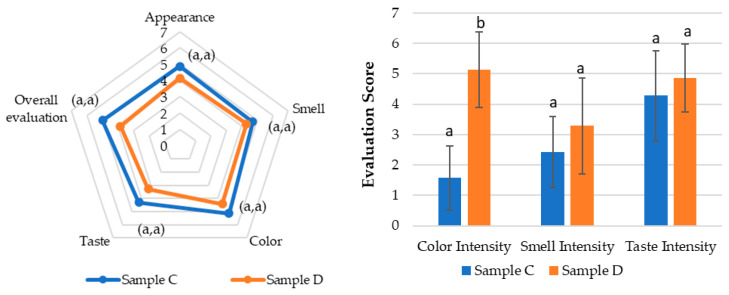
Sensory evaluation of cannabis samples. (**Left**) Radar chart showing mean scores for appearance, smell, color, taste, and overall assessment of final high-CBD MCT solutions samples C and D. (**Right**) Bar graph comparing color, smell, and taste intensity between samples C (SOMAÍ) and D (Tilray). Values represent mean scores given by the panel, with statistically significant differences (*p* < 0.05) indicated by letters for each attribute.

**Figure 9 pharmaceutics-17-00848-f009:**
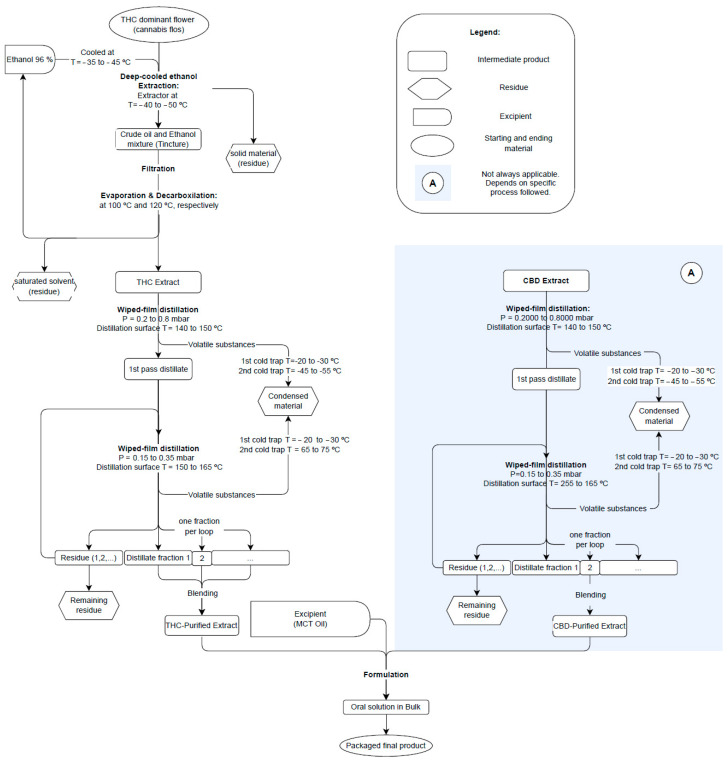
Flowchart of SOMAÍ’s process to yield high-THC or high-CBD herbal substances (purified soft extracts), highlighting key steps such as terpene extraction, deep-cooled ethanol extraction, purification to retain cannabinoids while removing impurities, and reintroduction of terpenes.

**Figure 10 pharmaceutics-17-00848-f010:**
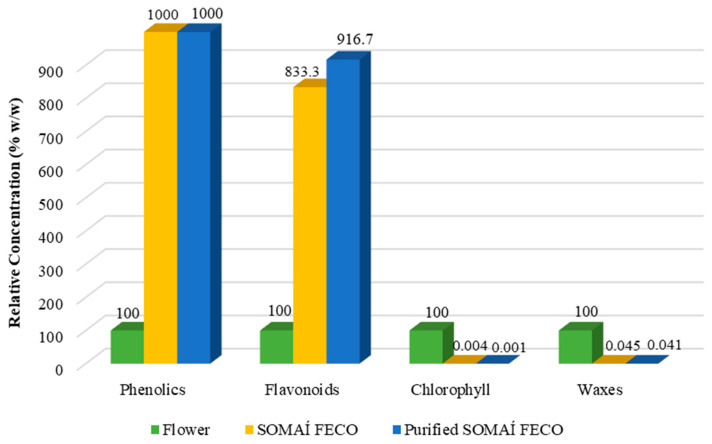
The relative concentration of phenolics, flavonoids, chlorophyll, and waxes in flower, crude full cannabis extract oil (SOMAÍ FECO), and purified SOMAÍ FECO. Values are expressed as a percentage (% *w*/*w*) normalized to the flower.

**Figure 11 pharmaceutics-17-00848-f011:**
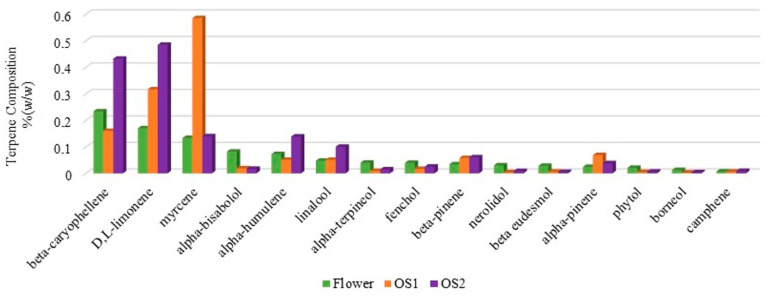
Evidence of conservative terpene composition (% *w*/*w*) in flower, oral solution 1 (OS1), and oral solution 2 (OS2) after reintroduction of terpenes, highlighting the conservation and restoration of major terpene components in the final formulations.

**Table 1 pharmaceutics-17-00848-t001:** Retention time (RT) and relative retention time (RRT) of cannabinoids.

Cannabinoid	RT	RRT
CBDV	2.76	0.35
CBDA	3.84	0.49
CBGA	4.14	0.53
CBG	4.40	0.56
CBD	4.56	0.58
THCV	4.62	0.59
THCVA	5.97	0.76
CBN	6.54	0.83
Δ9-THC	7.88	1.00
CBNA	8.03	1.02
Δ8-THC	8.15	1.03
CBL	9.05	1.15
CBC	9.61	1.22
Δ9-THCA	9.80	1.24
CBLA	10.89	1.38
CBCA	10.97	1.39

**Table 2 pharmaceutics-17-00848-t002:** Overview of the cannabinoid content (% *w*/*w*) at different FECO process stages (flower, crude extract, and purified extract) with values represented as the mean ± standard deviation of at least two replicates from 3 different batches.

Cannabinoids % (*w*/*w*)	Total THC	Total CBD	CBN	Total CBG	Other Cannabinoids	Total Cannabinoids
Flower	17.7 ± 1.9	0.11 ± 0.05	0.05 ± 0.04	1.5 ± 0.62	2.5 ± 0.50	21.9 ± 2.2
SOMAÍ FECO	73.2 ± 1.6	0.56 ± 0.20	1.1 ± 0.49	3.3 ± 1.3	8.5 ± 1.1	86.7 ± 1.6
Purified SOMAÍ FECO	83.4 ± 1.9	0.81 ± 0.18	1.2 ± 0.58	3.3 ± 1.3	7.3 ± 1.0	96.0 ± 2.2

**Table 3 pharmaceutics-17-00848-t003:** Terpene concentrations (% *w*/*w*) isolated during the extraction and purification process.

Terpenes	Recovered After Extraction
D,L-limonene	28.83
myrcene	18.50
beta-caryophellene	8.69
beta-pinene	3.36
linalool	2.71
fenchol	2.27
alpha-pinene	1.70
nerolidol	1.65
alpha-humulene	1.62
alpha-terpineol	1.32
L-Fenchone	0.55
terpinolene	0.53
camphene	0.49
borneol	0.44
geranyl acetate	0.19
gamma-terpinene	0.14
alpha-bisabolol	0.08
alpha-terpinene	0.08
Camphor	0.08
Cedrol	0.06

## Data Availability

The original contributions presented in this study are included in the article. Further inquiries can be directed to the corresponding author.

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
