# Peer review of "Process Development for GMP-Grade Full Extract Cannabis Oil: Towards Standardized Medicinal Use"

_pharmaceutics, 2025, doi:10.3390/pharmaceutics17070848_

Round 1

Reviewer 1 Report

Comments and Suggestions for Authors

1.The title of the paper needs to be readjusted, which doesn't look like the title of the research work.

2.What's the difference between “deep-cooled ethanol” and “subzero temperatures ethanol”? Please uniform them.

3.Please change the unit of mg/% with mg/g or wt.% in Table 2.

4.Please provide the supporting references for experimental methods, formulas and quantitative way of HPLC-PDA.

5.The current analytical methods appear unable to distinguish between flavonoids and phenolic compounds, as the former also contain phenolic hydroxyl groups.

6.The comma in Table 6 data should be a period.

7.Who is the main contributor for the bioactivity of oil product? How about the relationship between its dose and activity?

8.Error bars are missing in some figures.

9.Some references shoud be revised.

Author Response

#Comment 1

The title of the paper needs to be readjusted, which doesn't look like the title of the research work.

Response: the title of the paper has been improved to: " Process Development for GMP-Grade Full-Spectrum Cannabis Extract Oil: Towards Standardized Medicinal Use"

Dear Reviewer:

Please see the Attachment concerning all the other Comments which were very valuable, and the Responses and Changes introduced in the manuscript accordingly.

Best regards

The authors

Reviewer 2 Report

Comments and Suggestions for Authors
  1. The scheme in Figure 7 could use a large or bolder font for the reader’s convenience.
  2. In the opinion of the reviewer, the word “composition” in table 2, Figures 9 and 10 should be replaced with the word “content” in accordance with the text.

Author Response

Dear Reviewer, please find below the responsed to your most helpful comments:

#Comment 1

The scheme in Figure 7 could use a large or bolder font for the reader’s convenience

Response to Comment 1

Thank you for the suggestion. We have increased the size of Figure 7. The authors believe that increasing the overall size of Figure 7 enhances readability sufficiently. Since the article will be published in digital format, readers will also have the ability to zoom in for greater detail if desired. We hope this adjustment adequately addresses the concern, since we had difficulties to enlarge the font mantaining harmony of the layout.

#Comment 2

In the opinion of the reviewer, the word “composition” in table 2, Figures 9 and 10 should be replaced with the word “content” in accordance with the text. 

Response to Comment 2

Thank you for your valuable suggestion. We have revised the captions for Table 2, Figure 9, and Figure 10 (now updated to Table 3, Figure 8 and Figure 9) by replacing the word “composition” with “content” to ensure consistency with the terminology used in the main text and to more accurately reflect the quantitative nature of the data presented.

Best regards

The authors

Reviewer 3 Report

Comments and Suggestions for Authors

Lines 93-96 of the introduction: Why do they include results from their work in the introduction? That is incorrect.

Lines 139-140: The authors state that “The end of the decarboxylation reaction is monitored using a thermogravimetric analyzer (Sartorius MA160).” They should mention the technique used or describe how they monitored the reaction.

Line 203: The authors created a calibration curve for the HPLC analysis of Δ⁹-THCA, but they do not mention which standard they used for this.

Line 216: “Absorbance was recorded at 750 nm using a spectrophotometer.” They should mention the equipment used.

The authors mention that they quantified the terpene content by GC-MS, but they do not mention the standards they used for this, nor how they created the calibration curves. They also mention having developed the chromatographic technique, but they do not provide details. They should detail it.

According to the values ​​reported in Table 2, the concentrations of polyphenols and flavonoids do not change after the purification process. However, the authors mention in the introduction that “The task was to maintain all the medicinal qualities of the original plant chemovars [17,18] and remove undesirables with no defined therapeutic value, such as lipids, carbohydrates, chlorophyll, and polyphenols, in the gentlest possible way.” Therefore, the purification process they used was not effective in meeting this objective. In fact, in the discussion, the authors state that “Lipids, chlorophyll, and polyphenols will not be distilled and instead will travel to the waste of the process,” which is not true. Furthermore, according to the data presented in Table 3, terpenes are practically eliminated during the process (from a concentration of 1.31% wt. to 0.29% wt.). According to the stated objective, the purification process should preserve these compounds that have therapeutic action.

Why don't they report the THC and CBD values ​​in their acidic and neutral forms separately?

In the discussion of results, it is necessary to include studies conducted by other authors, both regarding the chemical composition, the extraction and purification process, and the sensory and acceptance evaluation. This will facilitate the comparison of the results obtained by the authors.

The authors state, "Throughout cannabis processing, non-cannabinoid compounds change, especially during the transition from flower to SOMAÍ FECO, as seen in Figure 10." This statement is incorrect, as evidenced by an analysis of the information in Table 2 and Figure 10 (which is also the same information repeated). The authors then state on lines 597-598, "while no changes were observed in phenolic content throughout the purification process," the opposite of what they stated previously.

The authors state that they concentrated the terpenes extracted during the purification process and then reintroduced them, allowing the final product to regain its characteristic flavor and aroma. However, the methodology does not specify how they achieved this concentration or how they reintroduced them (proportion, etc.).

Specific Comments on Tables and Figures

  • The information in Figures 2 and 4 is repeated in Figures 5 and 7, respectively. I suggest placing them only in Figures 5 and 7.
  • Figure 8 is unnecessary because the information it contains is the same as that reported in Table 1.
  • Figure 9 repeats information already presented in Table 1, and Figure 10 repeats the information presented in Table 2 and then repeats it in the discussion text. The authors should redesign the manuscript so that the info appears only once.
  • Figure 11 is completely irrelevant and again presents the same information as Table 2, Figure 10, and what appears in the text.
  • Figures 12 and 13 present the same information as Table 3.

Author Response

Dear Reviewer 3

Thank you very much for your valuable comments which provided a very deep brainstorming among the authors.

Please see the attachment with author´s responses for your kind consideration.

Best regards

The authors

Round 2

Reviewer 1 Report

Comments and Suggestions for Authors
  1. Full Spectrum Extract Cannabis Extract Oil should be abbreviated as FSECEO.
  2. When readers read the THC in keywords, they do not know what they are. Please ensure to provide the full names when all the abbreviations appear.
  3. Error bars are missing in some figures.
  4. please provide a special scheme for the operations and instruments in section 2.3 amd 2.4.
  5. please use three-line format for all the tables.
  6. there are two Figure 12.

Author Response

#Comment 1

Full Spectrum Extract Cannabis Extract Oil should be abbreviated as FSECEO.

Response to Comment 1

Thank you for your comment. We were also not happy with nomenclature, after trying to adapt to First Round 3 Reviewers comments . Indeed there is a redundant duplicated term: "Extract".  After a thorough review of current literature and industry standards, we have maintained the abbreviation FECO (Full Extract Cannabis Oil). This term is widely recognized and utilized within the field, compared to the proposed abbreviation FSECEO. Therefore, we have opted to retain FECO to ensure clarity and consistency in the academic approach, aligning it with industry market standards and nomenclature within our manuscript.

#Comment 2

When readers read the THC in keywords, they do not know what they are. Please ensure to provide the full names when all the abbreviations appear.

Response to Comment 2

Thank you for your suggestion. We have updated the keywords section in the manuscript accordingly.

#Comment 3

Error bars are missing in some figures.

Response to Comment 3

Dear Reviewer,

Thank you for your valuable and precise comments. We now provide a more complete clarification:

In Figure 9, the data are expressed as normalized relative percentages, with the inflorescence fixed at 100% as reference. Since normalization suppresses absolute variation and redistributes values proportionally across all detected monoterpenes, adding conventional error bars would be mathematically inconsistent and potentially misleading, as variability would no longer reflect true experimental uncertainty but rather transformed proportions. For transparency, we have now included the full set of raw (pre-normalization) values and replicate data in the Supplementary Material (new file added).

Regarding Figures 2 and 10, these results are non-statistical process profiles representing exact terpene contents obtained from a defined batch series (flower → semi-finished product → final product). These are single-batch, traceable, GMP-reported values obtained under standardized and validated conditions, without replicates or population-based sampling. Therefore, the values shown are deterministic within the precision of the analytical system (GC-FID), and no error estimation or statistical variability applies in this context.

Nevertheless, to ensure full transparency and support your point, we have included the instrumental raw data and batch documentation (including GC-FID reports and standard curves) as part of the Supplementary Material. We fully agree with your concern and believe this additional material addresses it rigorously.

Thank you once again for your constructive input, which helped us strengthen the methodological clarity of the manuscript.

#Comment 4

Please provide a special scheme for the operations and instruments in section 2.3 and 2.4.

Response to Comment 4

We thank the reviewer for the pertinent request.

Indeed, a detailed scheme illustrating the operations and instruments used in Sections 2.3 (High-THC Cannabis sativa L. Flower Extraction) and 2.4 (Purification Process) is currently part of an industrial process under intellectual property protection. The corresponding flowchart and operational details are integral to the subject matter of the patent application cited as reference [24]: 

Sassano, M. Cannabis Flower Extraction Method and Extract Formulations, European Patent Application No. EP24218305.1, Somaí PHARMACEUTICALS LIMITED, 09 December 2024.

This patent is currently under application revision phase following the issuance of the EPO Search Report. Consequently, we are temporarily unable to disclose the full schematic diagram to avoid compromising the novelty and scope of protection sought.

Nonetheless, to support transparency and reproducibility, we have provided all permissible methodological details within the manuscript and included instrumentation specifications where relevant, at LINES 166-179.

Upon completion of the patent process, we will be pleased to share the full schematic upon request or in a follow-up publication.

#Comment 5

Please use a three-line format for all the tables.

Response to Comment 5

Thank you for your requirement on further adjustments and clarifications. We have revised Table 1 to adhere to the three-line format as per your recommendation. Upon review, we believe that all other tables in the manuscript already conform to this format. 

#Comment 6

There are two Figure 12.

Response to Comment 6

Thank you for your observation. We acknowledge the duplication in the figure numbering. To resolve this, we have removed the original Figure 12B and retained only Figure 12A, which is now labeled as Figure 10. Additionally, the figure previously designated as Figure 11 has been relocated to the Results section to maintain clarity and consistency throughout the manuscript, and we moved the table to the supplementary materials. We trust these adjustments address your concern.

Best regards

Reviewer 3 Report

Comments and Suggestions for Authors

After a thorough review of the manuscript provided by the authors, I acknowledge that many of the comments and suggestions I made have been taken into account. However, some of them have been refuted. In this regard, I believe that comment 2 should be addressed and clarified, and it will be the editor's decision whether to address the remaining comments detailed below.

Comment 2:

Despite the explanation the authors added to the manuscript, it is not clear to what temperature they heated the sample to ensure that only the acid form of the cannabinoids was being decarboxylated, and that no other developmental reaction, for example, of cannabinoids or other cannabis compounds was occurring.

Comments 11 to 15

Despite the authors' explanation, I maintain that according to scientific standards, information should not be repeated in more than one format: if a graph is used, the same data should not be included in a table. In any case, the editor will have to make a decision on this matter.

Author Response

#Comment 2

Despite the explanation the authors added to the manuscript, it is not clear to what temperature they heated the sample to ensure that only the acid form of the cannabinoids was being decarboxylated, and that no other developmental reaction, for example, of cannabinoids or other cannabis compounds was occurring.

Response to Comment 2

The following information has been added at LINES 166-179 and 185-188.

Sartorius MA160 moisture analyzer, although being a nonspecific analytical method, is used as In Process Control (IPC), at the shopfloor, to monitor the end of the decarboxylation process, for it is quick and easy to use. A qualification process has been previously performed to make the parallel between the HPLC quantification of the THCA/THC and the analytical limit established for the moisture analyzer results (not more than 2.5%  at 120 ºC for a 2g sample for máx. 30 minutes).

The sample mass is continuously recorded under controlled heating conditions, and stabilization of the weight over time is used as an indirect indicator of the completion of decarboxylation. This approach is based on the assumption that the decarboxylation of acidic cannabinoids (e.g., THCA to THC) is associated with the release of CO₂, leading to a measurable mass loss.

The reaction was considered complete when no further significant mass loss (<0.001 g/min) was observed, but the final quantification and the results reported are obtained in final Quality Control Analysis by HPLC as further described in the article.”

# Comments 11 to 15

Despite the authors' explanation, I maintain that according to scientific standards, information should not be repeated in more than one format: if a graph is used, the same data should not be included in a table. In any case, the editor will have to make a decision on this matter.

Response to Comments 11 to 15

Dear Reviewer, thank you for your most valuable feedback. In response to your concerns regarding data presentation, we've adjusted to enhance clarity and align to best practices in scientific publishing by moving some of the tables to the supplementary materials for readers seeking precise data. We trust these revisions address your observations and contribute to the overall quality of the manuscript.

Concerning the comment on that no other developmental reaction, for example, of cannabinoids or other cannabis compounds was occurring, the analytical control through all phases of the process provides evidence that  waxes and chlorophyll reduced very significantly, while no changes were observed in total phenolic content,  while the cannabinoid full spectrum is accurately monitored, including degradation products after stability assays during 36 months like CBN and delta-8THC (without significant changes, out ofthe scope of this paper).